# Security Analysis of DBTRU Cryptosystem

**DOI:** 10.3390/e24101349

**Published:** 2022-09-23

**Authors:** Xiaofei Tong, Jingguo Bi, Yufei Duan, Lixiang Li, Licheng Wang

**Affiliations:** 1School of Cyberspace Security, Beijing University of Posts and Telecommunications, Beijing 100876, China; 2State Key Laboratory of Networking and Switching Technology, Beijing University of Posts and Telecommunications, Beijing 100876, China; 3State Key Laboratory of Cryptology, P.O. Box 5159, Beijing 100878, China

**Keywords:** DBTRU, NTRU, public key cryptosystem, linear algebra attack, lattice-based attacks

## Abstract

DBTRU was proposed by Thang and Binh in 2015. As a variant of NTRU, the integer polynomial ring is replaced by two binary truncated polynomial rings $GF\left(2\right)\left[x\right]/\left(x^{n}+1\right)$. DBTRU has some advantages over NTRU in terms of security and performance. In this paper, we propose a polynomial-time linear algebra attack against the DBTRU cryptosystem, which can break DBTRU for all recommended parameter choices. The paper shows that the plaintext can be achieved in less than 1 s via the linear algebra attack on a single PC.

## 1. Introduction

The Number Theory Research Unit (NTRU) cryptosystem as a public key cryptosystem was proposed by Hoffstein, Pipher, and Silverman in 1996 and published in 1998 [1]. It was standardized by IEEE in 2008 [2]. In 2020, NTRU entered the third round of submissions in the National Institute of Standards Technology (NIST) post-quantum cryptography standardization process. NTRU works on the integer polynomial ring $Z\left[x\right]/\left(x^{n}-1\right)$. The encryption and decryption procedures involve linear operations between ring elements. This characteristic gives NTRU a great advantage over Rivest, Shamir, Adleman (RSA) cryptosystem and elliptic curve cryptosystem (ECC) in terms of computational speed and key size. NTRU can be classified as post-quantum cryptography, and its security is based on the hardness of the shortest vector problem in a certain lattice. Compared with traditional public key algorithms, its research has been a hot spot in the field of public key cryptography. NTRU is widely used in e-commerce, communication, embedded systems, and portable devices [3,4].

Since 2002, cryptographers have been exploring the optimization of NTRU from the underlying mathematical structure in order to achieve a higher level of security or better performance. Banks et al. gave the non-invertible version in 2002 [5]. This extension can overcome the problem of finding “enough” invertible polynomials in small sets. In 2002, Gaborit et al. proposed CTRU [6], a NTRU-like cryptosystem that runs on $F_{2}\left[T\right]\left[X\right]/\left(x^{n}-1\right)$. CTRU can avoid the attacks based on the LLL algorithm. Vats proved that it is insecure under linear algebra attack in 2008 [7]. In 2005, Coglianese and Goi proposed MaTRU [8], which operates in the ring of *k* by *k* matrices $M_{k}\left(Z\right)\left[X\right]/\left(x^{n}-1\right)$. Compared to NTRU, MaTRU further improves system operation efficiency. In 2011, Malekian et al. adopted the unique mathematical structure of quaternion algebra to design the QTRU cryptosystem [9], in which non-commutativeness plays a key role in the system, and which further enhances the security of QTRU. In 2015, Yasuda et al. proposed a general NTRU cryptosystem based on group ring, called GR-NTRU [10]. They investigated the security and performance of the cryptosystem under different instance group rings by combining group representation theory. In 2017, Thakur et al. designed NTRU over spit quaternion algebra [11]; SQTRU can reduced the decryption failure due to a non-commutative algebraic structure. In 2018, Wang et al. presented a variant of NTRU with IND-CPA security named D-NTRU [12], which has higher encryption and decryption efficiency than NTRU. In 2008, Karbasi et al. established PairTRU working in the k×k matrix ring with pairwise entries of $k^{2}$ distinct polynomials in Z×Z [13]. PairTRU is more secure than NTRU under lattice based attack. In 2020, Hajaje et al. proposed PMTRU by combining the advantages of NTRU with MATRU [14]. PMTRU also improves the speed of encryption and decryption procedures.

DBTRU was proposed by Thang and Binh in 2015 [15]. The name DBTRU indicates the use of number theory and two binary truncated polynomial rings $GF\left(2\right)\left[x\right]/\left(x^{n}+1\right)$, ($n\in Z^{+}$). Because both algorithms for encryption and decryption of DBTRU are only simple modular polynomial operations, DBTRU is as fast as NTRU. Although the message-expansion factor in DBTRU is higher than that in NTRU, the keys of DBTRU are smaller under approximately the same level of security.

In this paper, we further analyze the security of DBTRU and propose a linear algebra attack that can break it for all recommended parameter choices to compare the security levels in NTRU. More precisely, we first explore a hidden linear relationship between the public keys and the secret keys and find the parameter constraints for plaintext and secret key security while guaranteeing correct decryption.

The rest of this paper is organized as follows. In Section 2, we briefly describe the DBTRU encryption scheme. In Section 3, we show how to recover the plaintext under the linear algebra attack. In Section 4, the experimental results of our attack are provided. We give the conclusions in Section 5.

## 2. The DBTRU System

We describe the DBTRU cryptosystem, as developed in [15], including notations, key generation, encryption, decryption, and decryption criteria.

### 2.1. Notations

This cryptosystem relies on two integer parameters *s*, *l* and four sets $B_{f}$, $B_{g}$, $B_{\phi }$, $B_{m}$ of polynomials with binary coefficients. In general, *s* is smaller than *l* and gcd(s,l)=1. Let $R=Z\left[x\right]/\left(x^{n}-1\right)$. The polynomial ring $GF\left(2\right)\left[x\right]/\left(x^{n}+1\right)$ is denoted by $R_{n}\left[x\right]$. DBTRU is working in $R_{s}\left[x\right]$ and $R_{l}\left[x\right]$. We write * for polynomial multiplication in $R_{n}\left[x\right]$, and let deg(f) denote the degree of $f\in R_{n}\left[x\right]$.

Let $d_{f}$, $d_{g}$, $d_{\phi }$, and $d_{m}$ denote the maximum degree and Hamming weight of *f*,*g*,ϕ, and *m*, respectively. We replace the definition $L(d_{1},d_{2})$ in NTRU with
$$B\left(d\right)=\{b\in R_{l}\left[x\right]|\deg\left(b\right)\leq d\}.$$

In addition, similar to DBTRU, we set the modular polynomials as $S=x^{s}+1$ and $L=x^{l}+1$.

### 2.2. Key Generation

During the process of key generation, Bob chooses two arbitrary positive integers *s* and *l* such that s<l, and sets $d_{f}=s-1$. In addition, Bob chooses an small positive integer $N_{f}$ and arbitrary $N_{f}$ polynomials $f_{i}\in B_{f}$ ($i\in [1,N_{f}]$), which are invertible in both $R_{s}\left[x\right]$ and $R_{l}\left[x\right]$. For each $f_{i}$, Bob computes $F_{i,s}\in R_{s}\left[x\right]$ and $F_{i,l}\in R_{l}\left[x\right]$, where $F_{i,s}*f_{i}\equiv 1\;\mathrm{mod}\;S$ and $F_{i,l}*f_{i}\equiv 1\;\mathrm{mod}\;L$. Then Bob computes
$$f=\prod_{i=1}^{N_{f}}f_{i},$$
and its two inverses
$$F_{s}=\prod_{i=1}^{N_{f}}F_{i,s},$$
and
$$F_{l}=\prod_{i=1}^{N_{f}}F_{i,l}.$$

Notice that $\deg\left(f\right)\leq N_{f}\cdot d_{f}$. Bob chooses a non-zero polynomial $g\in B_{g}$ and computes
$$h=g*F_{l}*S\;\mathrm{mod}\;L.$$

Bob keeps *f*,$f_{i}$, and $F_{s}$ as the private keys, publishing *h* as the public key.

### 2.3. Encryption and Decryption

Suppose Alice wants to send a *s*-bit message *m* to Bob. First, Alice randomly selects a non-zero polynomial $\phi_{0}\in B_{\phi }$, a small positive integer $N_{\phi }$, and arbitrary $N_{\phi }$ polynomials $\phi_{i}\in B_{\phi }|i\in \left[1,N_{\phi }\right]$. The ciphertext is given by
(1)$$e\equiv (\phi_{0}*h+S*\sum_{i=1}^{N_{\phi }}\phi_{i}+m)\;\mathrm{mod}\;L.$$

Alice then sends the *l*- bit ciphertext *e* to Bob. After receiving *e*, Bob computes
(2)a≡f∗emodL,
and recovers the message *m* by computing
$$m\equiv F_{s}*a\;\mathrm{mod}\;S.$$

### 2.4. Proof of Decryption

By inserting (1) into (2), there is
$$\begin{array}{cc} a & \equiv f*e\;\mathrm{mod}\;L\\ \; & \equiv f*(\phi_{0}*h+S*\sum_{i=1}^{N_{\phi }}\phi_{i}+m)\;\mathrm{mod}\;L\\ \; & \equiv (f*\phi_{0}*g*F_{l}*S+f*S*\sum_{i=1}^{N_{\phi }}\phi_{i}+f*m)\;\mathrm{mod}\;L\\ \; & \equiv (\left(\phi_{0}*g+f*\sum_{i=1}^{N_{\phi }}\phi_{i}\right)*S+f*m)\;\mathrm{mod}\;L. \end{array}$$

Hence, $F_{s}*a\;\mathrm{mod}\;S=F_{s}*f*m\;\mathrm{mod}\;S$. Thereby,
$$m\equiv F_{s}*a\;\mathrm{mod}\;S.$$

### 2.5. Decryption Criteria

It is proved that
$$\deg\left(\phi_{0}*g*S\right)\leq d_{\phi }+d_{g}+s,$$
$$\deg\left(f*S*\sum_{i=1}^{N_{\phi }}\phi_{i}\right)\leq N_{f}\cdot d_{f}+s+d_{\phi },$$
and
$$\deg\left(f*m\right)\leq d_{f}+d_{m},$$
if $d_{g}$ satisfies
$$d_{g}<N_{f}\cdot d_{f},$$
then
$$\deg\left(a\right)\leq N_{f}\cdot d_{f}+s+d_{\phi },$$
Thereby, to ensure successful decryption, it is necessary that
$$l>\max\left(\deg a\right)=N_{f}\cdot d_{f}+d_{\phi }+s.$$

## 3. Security Analysis

In this section, we describe the details of our attack on a DBTRU cryptosystem. First, we show that there is a hidden linear relationship between the public keys and the random non-zero polynomial in the encryption phase. Second, we construct a linear system of equations with the unknown random non-zero polynomial and then recover the plaintext message after we obtain the random non-zero polynomial. Finally, we present the whole algorithm of our attack.

### 3.1. The Hidden Linear Relationship

**Theorem** **1.**
*As described in the DBTRU cryptosystem, let $S=x^{s}+1$ and $L=x^{l}+1$, where s<l. Let $\phi_{i}\in B_{\phi }$ ($i=0,1\cdots ,N_{\phi }$) be some randomly chosen polynomials with $\phi_{0}\neq 0$. For the ciphertext*

$$e=(\phi_{0}*h+S*\sum_{i=1}^{N_{\phi }}\phi_{i}+m)\;\mathrm{mod}\;L,$$

*if $l\geq s+2d_{\phi }+2$, then the part of coefficients of e, namely, $e_{s+d_{\phi }+1},\cdots ,e_{l-1}$ are equal to the coefficients of $\phi_{0}*h\;\mathrm{mod}\;L$ with the same degree.*


**Proof** **of Theorem 1**As noted above, the ciphertext is calculated by
$$e=(\phi_{0}*h+S*\sum_{i=1}^{N_{\phi }}\phi_{i}+m)\;\mathrm{mod}\;L,$$
and we can write *e* as
$$e=\sum_{i=0}^{l-1}e_{i}x^{i},$$
where $e_{i}\in GF\left(2\right)$ (i=0,1,⋯,l−1). We assume
$$\phi_{0}=\alpha_{0}+\alpha_{1}x+\cdots +\alpha_{d_{\phi }}x^{d_{\phi }},$$
where $\alpha_{i}\in GF\left(2\right)$ ($i=0,1,\cdots ,d_{\phi }$). In addition,
$$h=h_{0}+h_{1}x+\cdots +h_{l-1}x^{l-1},$$
with $h_{j}\in GF\left(2\right)$ (j=0,1,⋯,l−1).We have
$$\deg\left(S*\sum_{i=1}^{N_{\phi }}\phi_{i}+m\right)\leq s+d_{\phi }.$$
Now considering the maximum degree of components of $\phi_{0}*h$, we have
$$\deg\left(\phi_{0}*h\right)\leq d_{\phi }+d_{h}=d_{\phi }+l-1.$$
From the precise analysis above, we have only part of the coefficients of *e* related to the $\phi_{0}*h$, $S*\sum_{i=1}^{N_{\phi }}\phi_{i}$ and *m*. More specifically, only the coefficients $e_{0},e_{1},\cdots ,e_{d_{\phi }-1}$ are affected by the modulo *L*, and $e_{s+d_{\phi }+1},\cdots ,e_{l-1}$ are just equal to the coefficients of $\phi_{0}*h\;\mathrm{mod}\;L$ with the same degree.    □

From Theorem 1, we can see that the key to breaking DBTRU lies in the irrationality of the ciphertext structure. In each encryption process, we can construct the following linear equation system through the partial coefficients $e_{k}=\sum_{i+j=k}\alpha_{i}\cdot h_{j}\;\left(s+d_{\phi }+1\leq k\leq l-1\right)$ of the ciphertext *e*, we have
(3)$$\left\{\begin{array}{c} h_{l-1}\alpha_{0}+h_{l-2}\alpha_{1}+\cdots +h_{l-d_{\phi }-1}\alpha_{d_{\phi }}=e_{l-1}\\ h_{l-2}\alpha_{0}+h_{l-3}\alpha_{1}+\cdots +h_{l-d_{\phi }-2}\alpha_{d_{\phi }}=e_{l-2}\\ \cdots \cdots \\ h_{s+d_{\phi }+1}\alpha_{0}+h_{s+d_{\phi }}\alpha_{1}+\cdots +h_{s+1}\alpha_{d_{\phi }}=e_{s+d_{\phi }+1}. \end{array}\right.$$

We denote the coefficient matrix of Equation (Equation 3) as
$$A=\left[\begin{array}{cccc} h_{l-1} & h_{l-2} & \cdots & h_{l-d_{\phi }-1}\\ h_{l-2} & h_{l-3} & \cdots & h_{l-d_{\phi }-2}\\ \vdots & \vdots & \ddots & \vdots \\ h_{s+d_{\phi }+1} & h_{s+d_{\phi }} & \cdots & h_{s+1} \end{array}\right],$$
where the elements of the matrix are the coefficients of the public key *h*.

In Equation (Equation 3), the number of variables is $d_{\phi }+1$, and the number of equations is $l-s-d_{\phi }-1$. Let
$$l\geq s+2d_{\phi }+2,$$
we have that the number of equations is greater than or equal to the number of variables. In this case, the system of equations in (Equation 3) has a unique solution. Therefore, plaintext and secret polynomial $\phi_{0}$ will be secure if
$$l<s+2d_{\phi }+2.$$
We will present how to recover the unique solution $\phi_{0}$ in the next subsection.

**Remark** **1.**
*In the DBTRU cryptosystem, the authors also proposed an assessment of the algebraic attack on this scheme. The main problem with their security analysis is that they paid attention to too many unknown polynomials. Here, we discover the hidden linear relationship between the public keys and the random non-zero polynomial by careful analysis.*


### 3.2. Recover the Non-Zero Polynomial $\phi_{0}$

To recover the polynomial $\phi_{0}$, we need to analyze the solutions of Equation (Equation 3). As long as the rank of matrix *A* defined above is equal to *n*, then Equation (Equation 3) should have only one solution, namely, the polynomial $\phi_{0}$. To analyze the rank of matrix A, we cite the following result, which is Theorem 2 of [16].

**Lemma** **1.**
*Let N be a positive integer. Let $p_{1},\cdots ,p_{l}$ be the distinct prime factors of N. Consider the ring of n×n matrices with entries in $Z_{N}$. Then the proportion of invertible matrices (i.e., with determinant coprime to N) is equal to :*

$$\prod_{i=1}^{l}\prod_{k=1}^{n}\left(1-p_{i}^{-k}\right).$$



Applying Lemma 1, we have the following Corollary.

**Corollary** **1.**
*Let p be a prime integer and t≥0 be an integer. Let $M_{(n+t)\times n}\left(Z_{p}\right)$ denote the ring consisting of (n+t)×n matrices with entries in $Z_{p}$. The probability of having at least one n×n invertible matrix in $M_{(n+t)\times n}\left(Z_{p}\right)$ is*

$$1-{\left(1-\prod_{k=1}^{n}\left(1-p^{-k}\right)\right)}^{\left(\begin{array}{c} n+t\\ n \end{array}\right)}.$$



**Proof** (Proof of Corollary 1).When setting N=p in Lemma 1, we have that the probability of having a irreversible n×n matrix with entries in $Z_{p}$ is
$$1-\prod_{k=1}^{n}\left(1-p^{-k}\right).$$Then, when we choose matrices from $M_{(n+t)\times n}\left(Z_{p}\right)$, the probability that all the n×n matrices are irreversible is
$${\left(1-\prod_{k=1}^{n}\left(1-p^{-k}\right)\right)}^{\left(\begin{array}{c} n+t\\ n \end{array}\right)}.$$
Based on the above analysis, we can deduce the result in our corollary.    □

Table 1 shows the probability of having at least one n×n invertible matrix in $M_{(n+t)\times n}\left(Z_{p}\right)$.

**Remark** **1.**
*From Table 1, we can see that even for p = 2, we only need to choose 3 times or more from $M_{(n+t)\times n}\left(Z_{p}\right)$; then we can get a invertible n×n matrix with a probability close to 1.*


Finally, after obtaining $\phi_{0}$, one can recover the message *m* by calculating
$$m\equiv (e-\phi_{0}*h)\;\mathrm{mod}\;S.$$

Here, we propose our whole attack as follows Algorithm 1   
**Algorithm 1:** Main strategy of this attackInput: $e_{k}=\sum_{i+j=k}\phi_{i}^{\prime }\cdot h_{j}\;\left(s+d_{\phi }+1\leq k\leq l-1\right)$ .1.Choose $d_{\phi }+1$ equations from the input system of linear equations, and denote its coefficient matrix as *A*.2.Determine whether detA is equal to be zero.3.If the detA≠0, apply Gaussian elimination to get the solution $a=(a_{0},a_{1},\cdots ,a_{d_{\phi }})$ of the selected systems of equations in Step 1.4.Else, then reselect $d_{\phi }+1$ equations, and go back to Step 2, until we find a system of equations for which its coefficient matrix is invertible.5.For all equations entered, check if $a=(a_{0},a_{1},\cdots ,a_{d_{\phi }})$ is a solution to each equation. If so, then we claim to have the target polynomial $\phi_{0}$.6.Compute $(e-\phi_{0}*h)\;\mathrm{mod}\;S$.Output: The s−bit plaintext message *m*.

## 4. Experiments Results

In DBTRU, the authors concluded that as a variant of NTRU, DBTRU has advantages in both security and performance comparison with NTRU, as shown in Table 2, Table 3 and Table 4, respectively.

Here, we use Sage Math to complete our experiments. First, we give the probability of encountering an invertible matrix when selecting multiple times under 10,000 sets of data in Table 5.

From Table 5, the experiment data validate Remark 2.

Next, we give the total running time of breaking the DBTRU cryptosystem under 10,000 sets of data in Table 6.

From Table 6, the results show that for the three parameter choices recommended in the DBTRU cryptosystem, our proposed linear algebra attack can recover the plaintext within 1 s.

## 5. Conclusions

The DBTRU cryptosystem is a binary analogue of NTRU. It was claimed in [15] that DBTRU has some important security and performance advantages over NTRU. For instance, at nearly the same level of security, DBTRU always has smaller keys. In this paper, we propose a linear algebra attack that breaks DBTRU by exploiting the secret linear relationship between public keys and secret keys. The linear algebra attack is practical on all three settings of recommended parameters, and the plaintext can be achieved in less than 1 s on a single PC. Our work may provide a new method of security analysis for NTRU variants or other cipher schemes.

Further research direction could be the fusion of NTRU with more complex algebraic structures, such as non-commutative algebras, to enhance the security of NTRU-like cryptosystems.

## Figures and Tables

**Table 1 entropy-24-01349-t001:** The probability of at least one n×n invertible matrix in $M_{(n+t)\times n}\left(Z_{p}\right)$, with p=2.

	*t*	t=0	t=1	t=2	t=3
*n*					
n=28		0.28879	0.99995	1.00000	1.00000
n=45		0.28879	1.00000	1.00000	1.00000
n=148		0.28879	1.00000	1.00000	1.00000

**Table 2 entropy-24-01349-t002:** Comparison in moderate security mode of NTRU.

Moderate Security	NTRU	DBTRU
Basic parameters	$\begin{array}{c} c\left(N,p,q,d_{f},d_{g},d\right)=\;\left(107,3,64,15,12,5\right) \end{array}$	$\begin{array}{c} c\left(s,l,d_{\phi },d_{g},N_{f},N_{\phi }\right)=\;\left(37,197,27,105,3,4\right) \end{array}$
$S_{m}$	$2^{26.5}$	$2^{51.21}$
$S_{k}$	$2^{50}$	$2^{51.71}$
Public key (bits)	642	197
Private key (bits)	340	222
Message-expansion	3.78	5.32

**Table 3 entropy-24-01349-t003:** Comparison in high security mode of NTRU.

High Security	NTRU	DBTRU
Basic parameters	$\begin{array}{c} c\left(N,p,q,d_{f},d_{g},d\right)=\;\left(167,3,128,61,20,18\right) \end{array}$	$\begin{array}{c} c\left(s,l,d_{\phi },d_{g},N_{f},N_{\phi }\right)=\;\left(59,293,44,120,3,4\right) \end{array}$
$S_{m}$	$2^{77.55}$	$2^{85.71}$
$S_{k}$	$2^{82.9}$	$2^{85.71}$
Public key (bits)	1169	293
Private key (bits)	530	354
Message-expansion	4.23	4.97

**Table 4 entropy-24-01349-t004:** Comparison in highest security mode of NTRU.

Highest Security	NTRU	DBTRU
Basic parameters	$\begin{array}{c} c\left(N,p,q,d_{f},d_{g},d\right)=\;\left(503,3,256,216,72,55\right) \end{array}$	$\begin{array}{c} c\left(s,l,d_{\phi },d_{g},N_{f},N_{\phi }\right)=\;\left(197,1019,147,500,3,4\right) \end{array}$
$S_{m}$	$2^{170}$	$2^{292.70}$
$S_{k}$	$2^{285}$	$2^{292.71}$
Public key (bits)	4024	1019
Private key (bits)	1595	1182
Message-expansion	5.05	5.17

**Table 5 entropy-24-01349-t005:** The probability of having an invertible matrix.

Parameters	Once	Twice	Three Times
$\begin{array}{c} c\left(s,l,d_{\phi },d_{g},N_{f},N_{\phi }\right)=\;\left(37,197,27,105,3,4\right) \end{array}$	0.2987	1	1
$\begin{array}{c} c\left(s,l,d_{\phi },d_{g},N_{f},N_{\phi }\right)=\;\left(59,293,44,120,3,4\right) \end{array}$	0.2957	1	1
$\begin{array}{c} c\left(s,l,d_{\phi },d_{g},N_{f},N_{\phi }\right)=\;\left(197,1019,147,500,3,4\right) \end{array}$	0.3033	1	1

**Table 6 entropy-24-01349-t006:** The running time for breaking DBTRU.

Parameters	The Number of Equations	The Number of Variables	Running Time (Sec)
$\begin{array}{c} c\left(s,l,d_{\phi },d_{g},N_{f},N_{\phi }\right)=\;\left(37,197,27,105,3,4\right) \end{array}$	132	28	15.7352
$\begin{array}{c} c\left(s,l,d_{\phi },d_{g},N_{f},N_{\phi }\right)=\;\left(59,293,44,120,3,4\right) \end{array}$	189	45	23.6364
$\begin{array}{c} c\left(s,l,d_{\phi },d_{g},N_{f},N_{\phi }\right)=\;\left(197,1019,147,500,3,4\right) \end{array}$	674	148	128.0634

## Data Availability

Not applicable.

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
