# Peer review of "Security Analysis of DBTRU Cryptosystem"

_entropy, 2022, doi:10.3390/e24101349_

Round 1
Reviewer 1 Report (New Reviewer)
This paper demonstrates an attack for the DBTRU proposed in a publication back in 2015.
I am not confident that the contributions of the paper justifies a publication in a journal. The attack is targeting a specific cryptographic algorithm named DBTRU published in a conference venue some years ago, and has not attracted much citations from the scientific community (Google scholar reports 2 citations).
Besides, the paper does not provide contributions beyond the attack. Frequently, in similar papers which a vulnerability has been discovered, the authors propose a remediation to secure the algorithm. The authors here simply present the attack, which makes the scope of the paper too narrow to justify a publication in a journal.
Also, the authors must improve their writing style. They mention that "We show that the security of DBTRU is overestimated". The authors do not show this. Instead, they show that the security of DBTRU is totally broken and there is no use of it as they were able to recover the plaintext message. Another example is the use of "3.1. Fatal Linear Relationship". The authors should avoid using words such as Fatal!
Author Response
Please see the attachment.

Reviewer 2 Report (New Reviewer)
Journal: Entropy
Manuscript ID: entropy-1845569
Type of manuscript: Article
TITLE: “Security Analysis of DBTRU Cryptosystem”
Authors: iaofei Tong , Jingguo Bi , Yufei Duan , Lixiang Li , Licheng Wang
Dear Editor,
This work focuses on using linear algebra under the parameter choices given in DBTRU, to secure a further analyze the security of the scheme.
The presentation and exhibition of work are well commented in the text and the bibliography describes well the background on the subject matter and the contributions of the research group. However, the emphasis on the novelty of this work is required.
This work fulfills, in my opinion, the requirements to be published with minor revision.
Here are some detailed comments on the manuscript:
- In the abstract and conclusion, the authors are required to explain the novelty of their work and the improvements compared to the previous works.
- In the article structure, there are some acronyms that need to be defined.
- The literature of the introduction requires the inclusion of some recent references.
- In section 2, the authors have introduced the notations in the DBTRU cryptosystem, however, they do not mention any references for all subsections.
- In section 4, the experimental results have to be explained and clarified for readers, not just the inclusion of the tables.
Best regards,
Reviewer
Round 2
Reviewer 1 Report (New Reviewer)
The authors have improved the paper significantly.
This manuscript is a resubmission of an earlier submission. The following is a list of the peer review reports and author responses from that submission.
Round 1
Reviewer 1 Report
Summary:
This paper proposes a linear algebra attack against DBTRU cryptosystem.
Strengths:
- The analysis of DBTRU is good.
Weaknesses:
- The proposed attack is not practical. Specifically, the attack only works based on the assumption that \phi_i's are **fixed** in each time of encryption, which is not true in DBTRU.
If Alice (re)samples \phi_i's in each time of encryption, the proposed schema cannot extract the coefficient matrix A from equation (3). The attack will fail accordingly.
Due to this fatal error in this paper, I am sorry that I cannot recommend accepting it.
- Typos and minor mistakes.
Line 22: “Although CTRU can avoid the lattice based attacks and Chinese Remainder Theorem.” The Chinese Remainder Theorem based attacks in lattice? "Avoid Chinese Remainder Theorem" is wired.
Line 23: "Although, ... but Vats proved ..." -> "Vat proved".
Line 35-36: "Although ..., but ..."
Line 52: "The polynomial ring ... denoted by ..." -> "is denoted by"
Line 53: "let ... denotes ..." -> "Let ... denote ..."
Line 56: "In the key generation process." is not a complete sentence.
Line 64, the brackets in the description of \phi_i seem missing.
Line 81: "coefficients matrix: -> "coefficient matrix"
Reviewer 2 Report
The article presents an analysis of the security level of a cryptographic system.
The statement in the conclusions: "DBTRU is completely insecure" is at least inadequate, once this system is used in practice, and the analysis was made for selected cases in the sense of highlighting the limitations of an encryption system and not its lack of applicability.